# Molecular analysis for ovarian cancer detection in patient-friendly samples
Birgit M. M. Wever [1,2], Mirte Schaafsma [1,2,3], Maaike C. G. Bleeker[1,2], Yara van den Burgt[1,2], Rianne van den Helder[1,2,3], Christianne A. R. Lok[3], Frederike Dijk[2,4], Ymke van der Pol[1,2], Florent Mouliere [1,2], Norbert Moldovan[1,2], Nienke E. van Trommel[3] & Renske D. M. Steenbergen [1,2] ✉

## Abstract

**Background** High ovarian cancer mortality rates motivate the development of effective and patient-friendly diagnostics. Here, we explored the potential of molecular testing in patient-friendly samples for ovarian cancer detection.

**Methods** Home-collected urine, cervicovaginal self-samples, and clinician-taken cervical scrapes were prospectively collected from 54 patients diagnosed with a highly suspicious ovarian mass (benign $n = 25$, malignant $n = 29$). All samples were tested for nine methylation markers, using quantitative methylation-specific PCRs that were verified on ovarian tissue samples, and compared to non-paired patient-friendly samples of 110 age-matched healthy controls. Copy number analysis was performed on a subset of urine samples of ovarian cancer patients by shallow whole-genome sequencing.

**Results** Three methylation markers are significantly elevated in full void urine of ovarian cancer patients as compared to healthy controls ($C2CD4D$, $P = 0.008$; $CDO1$, $P = 0.022$; $MAL$, $P = 0.008$), of which two are also discriminatory in cervical scrapes ($C2CD4D$, $P = 0.001$; $CDO1$, $P = 0.004$). When comparing benign and malignant ovarian masses, $GHSR$ shows significantly elevated methylation levels in the urine sediment of ovarian cancer patients ($P = 0.024$). Other methylation markers demonstrate comparably high methylation levels in benign and malignant ovarian masses. Cervicovaginal self-samples show no elevated methylation levels in patients with ovarian masses as compared to healthy controls. Copy number changes are identified in 4 out of 23 urine samples of ovarian cancer patients.

**Conclusions** Our study reveals increased methylation levels of ovarian cancer-associated genes and copy number aberrations in the urine of ovarian cancer patients. Our findings support continued research into urine biomarkers for ovarian cancer detection and highlight the importance of including benign ovarian masses in future studies to develop a clinically useful test.

## Plain language summary

Ovarian cancer is often found late with limited treatment options. Currently, it is difficult to diagnose ovarian cancer correctly and no recommended early detection or screening methods exist. Our aim was to explore the use of DNA-based tests in patient-friendly samples for ovarian cancer detection. Patient-friendly samples are patient materials that can be collected from home without pain or discomfort, such as self-collected vaginal swabs and urine. Using DNA-based tests, we found that urine of women with ovarian cancer contains ovarian cancer-associated signals. Our findings encourage further development of a potential urine test for ovarian cancer detection. This approach could aid early detection and guide women with ovarian masses to appropriate specialist care.

Ovarian cancer is the most lethal gynecological cancer worldwide, accounting for 207.252 deaths in 2020[1]. Due to non-specific or absence of symptoms at an early stage, patients typically present at a late-stage when prognosis is poor[2]. Five-year overall survival rates sharply decrease with higher stage at diagnosis, with 92% survival in early-stage disease compared to only 29% in late-stage disease[3]. High mortality rates prioritize the development of novel diagnostic approaches for ovarian cancer. Although more ovarian cancer patients were diagnosed at an earlier stage with

[1]Amsterdam UMC, location Vrije Universiteit Amsterdam, Department of Pathology, Amsterdam, The Netherlands. [2]Cancer Center Amsterdam, Imaging and Biomarkers, Amsterdam, The Netherlands. [3]Antoni van Leeuwenhoek/Netherlands Cancer Institute, Department of Gynecologic Oncology, Center of Gynecologic Oncology Amsterdam, Amsterdam, The Netherlands. [4]Amsterdam UMC, location University of Amsterdam, Department of Pathology, Amsterdam, The Netherlands. ✉e-mail: r.steenbergen@amsterdamumc.nl

screening strategies using conventional imaging and/or serum biomarkers (e.g., CA-125), this did not translate into reduced overall cancer-specific mortality in general populations[4,5]. In fact, the majority of ovarian cancers were not detected during or after the trial. A more accurate and easily accessible test could potentially overcome this problem.

Testing for ovarian cancer using biomarkers related to carcinogenesis could offer such an accurate test. DNA methylation-mediated silencing of tumor suppressor genes occurs early in cancer development and is therefore promising to detect cancer at an early stage[6]. Methylation analysis in urine, cervicovaginal self-samples, and clinician-taken cervical scrapes has already been proven to allow reliable detection of cervical[7,8] and endometrial cancer[9,10]. In urine, even signals of non-urogenital cancers, including colorectal[11] and lung cancer[12,13], are detectable by methylation testing. The measurement of somatic mutations, aneuploidy, or DNA methylation in clinician-taken cervical scrapes or blood demonstrated the high potential of molecular-based diagnostic tests for ovarian cancer[14–17]. However, these molecular changes have not been investigated in home-collected urine and cervicovaginal self-samples of ovarian cancer patients.

Urine and cervicovaginal self-samples are defined as patient-friendly sample types as both sampling strategies allow home collection without pain or discomfort. Experiences with these sampling strategies have been evaluated the context of cervical cancer screening, showing that both were considered acceptable and preferred by the majority of women[18–20]. Moreover, molecular testing in self-collected material could offer a cost-effective alternative to detect ovarian cancer, as assessed previously for cervical and endometrial cancer diagnostics[21,22].

In this study, we explore the potential of molecular testing in home-collected urine and cervicovaginal self-samples, and clinician-taken cervical scrapes for ovarian cancer detection. Methylation markers considered suitable for the detection of ovarian cancer include a combination of markers described in studies on cervical and endometrial cancer detection in patient-friendly sample types (*GALR1, GHSR, MAL, PRDM14, SST,* and *ZIC1*[10,23–25]), and ovarian cancer detection in cervical scrapes and plasma (*C2CD4D, CDO1, NRN1*[17,26–28]). In addition, the analysis of somatic copy number aberrations (SCNA) and fragmentation patterns is performed using shallow whole-genome sequencing on a subset of the samples to verify the presence of ovarian cancer-derived DNA in urine. Our data shows elevated methylation levels of a subset of markers and SCNA in home-collected urine samples of ovarian cancer patients. Yet, while urine offers an attractive sample type, similarly high methylation levels in urine of benign cases present a challenge in creating a clinically useful test.

## Methods
### Study population
This study prospectively included patients with a highly suspicious ovarian mass according to current triage methods (>40% risk of malignancy using the IOTA adnex model)[29,30]. Paired samples (i.e., urine, cervicovaginal self-samples, and clinician-taken cervical scrapes) were consecutively collected within the SOLUTION1 study, between July 2018 and September 2022, at the Antoni van Leeuwenhoek Hospital, Amsterdam, The Netherlands. Samples were collected from patients who underwent pelvic surgery with post-operatively confirmed ovarian cancer of any stage and histological subtype, and patients with a benign ovarian mass who were referred to a highly specialized tertiary oncology unit for further assessment. Patients scheduled for pelvic surgery, involving exploratory laparotomy to determine the origin of their ovarian mass or cytoreductive surgery, were asked to collect samples prior to surgery. Patients who could not collect cytological or urine samples prior to surgery were excluded from participation. Patients diagnosed with a borderline tumor were also excluded to focus on the most distinct tumor types in this exploratory stage (i.e., benign and malignant ovarian masses). Patients were included in the study regardless of whether all three paired sample types were available or not. For example, if a cervical scrape was not collected, the urine and self-sample of this patient were still analyzed and included.

Control urine samples were obtained from the URIC biobank, including healthy women without any prior cancer diagnosis within the last 5 years. Control cervicovaginal self-samples and cervical scrapes were retrieved from leftover material of the Dutch national cervical cancer screening program. Healthy control samples were within the same age range as women diagnosed with an ovarian mass, and all tested negative for high-risk human papillomavirus (HPV). Information on prior benign gynecological disease and menopausal status was not documented for healthy control women. Yet, the majority of women were most likely post-menopausal with 93% of healthy control women aged over 50 years.

To verify the discriminatory power of the methylation assays and concordance of copy number profiles, formalin-fixed paraffin-embedded (FFPE) and fresh frozen high-grade serous ovarian cancer (HGSOC) tissue samples were retrieved from the Pathology archives of Amsterdam UMC, locations AMC and VUmc, Amsterdam, The Netherlands. FFPE normal fallopian tube tissues were collected from patients undergoing a hysterectomy for the treatment of benign endometrial conditions.

### Ethics statement
Ethical approval was obtained by the Medical Ethical Committee of the VU University Medical Center for the use of samples collected within the SOLUTION1 study (METc: 2016.213, Trial registration ID: NL56664.029.16), samples stored in the URIC biobank (TcB 2018.657), and samples archived in the biobank containing leftover material of the Dutch national cervical cancer screening program (TcB 2020.245). Women participating in the screening program were informed that their residual cervical sample could be used for anonymized research and had the opportunity to opt-out. Only leftover material of women that did not opt-out was used. All study participants were 18 years or older and signed informed consent before sample collection. The Code of Conduct for Responsible Use of Left-over Material of the Dutch Federation of Biomedical Scientific Societies was adhered for the use of tissue specimens.

### Sample collection, processing, DNA extraction, and bisulfite modification
The sample collection, processing, DNA extraction, and bisulfite modification procedures were carried out as described previously for cervical[8,31] and endometrial cancer[10,24]. A detailed description is provided in the Supplemental Methods. Briefly, urine and cervicovaginal self-samples were collected at home and clinician-taken cervical scrapes were collected before surgery. Urine was centrifuged and separated into two fractions: the urine supernatant and the urine sediment. Both fractions and the remaining full void urine were stored for further analysis. Following DNA extraction, up to 250 ng of DNA was subjected to bisulfite modification.

### DNA methylation analysis by quantitative methylation-specific PCR
Methylation levels of the *C2CD4D* (gene-ID: 100191040), *CDO1* (gene-ID: 1036), *GALR1* (gene-ID: 2587), *GHSR* (gene-ID: 2693), *MAL* (gene-ID: 4118), *NRN1* (gene-ID: 51299), *PRDM14* (gene-ID: 63978), *SST* (gene-ID: 6750), and *ZIC1* (gene-ID: 7545) genes were measured by quantitative methylation-specific polymerase chain reactions (qMSP). Methylation markers were multiplexed to assess the methylation levels of three genes (1: *GHSR/SST/ZIC1*, 2: *CDO1/MAL/PRDM14*, 3: *C2CD4D/GALR1/NRN1*) and a reference gene (*ACTB*, gene-ID: 60) within the same reaction. Methylation analysis of *CDO1, GALR1, GHSR, MAL, SST, PRDM14*, and *ZIC1* was performed as described previously[10,23,24] with a shortened amplicon size of *ACTB, MAL* and *ZIC1* to facilitate methylation detection in fragmented urinary DNA. Assays targeting *C2CD4D* and *NRN1* were designed based on gene loci discovered and validated by others[17,26]. Primer and probe information is provided in Supplemental Table 1. Reaction conditions, instrument identifications, and thermocycling parameters are described in the Supplemental Methods. Double-stranded gBlocks™ Gene Fragments (Integrated DNA Technologies) containing the target amplicons and H$_2$O were taken along in each run as positive and negative control,

respectively. Sample quality and sufficient input was ensured by excluding samples with a *ACTB* quantification cycle (Cq) ≥ 32. Methylation levels were calculated relative to *ACTB* levels by the comparative Cq method: $2^{-(Cq\ marker\ –\ Cq\ ACTB)} \times 100$[32].

All qMSP assays were designed, multiplexed and optimized according to parameters described earlier[33]. Target specificity was validated in silico (BLAST). Correct amplicon size was verified by agarose gel electrophoresis. Analytical validation was performed using a dilution series of bisulfite-treated methylated DNA from the SiHa cell line (100, 50, 10, 5, 1, 0.5%) within the range of 20–0.1 ng (Supplemental Table 2). The discriminatory power of each assay was verified by comparing methylation marker levels in tissue samples of ovarian cancer patients with those measured in normal fallopian tube tissue.

### Shallow whole-genome sequencing
Urine cell-free DNA (cfDNA) extracted from urine supernatant samples of ovarian cancer patients was further characterized by shallow whole-genome sequencing (~1× coverage). The cfDNA was quantified and analyzed using a Cell-free DNA ScreenTape assay of the Agilent 4200 TapeStation System (Agilent) for quality control before sequencing. Sequencing libraries of the first pilot series of urine supernatant DNA were prepared using the ThruPLEX Plasma-seq Kit (Takara Bio, Mountain View, CA, USA) for whole-genome sequencing according to the manufacturer's instructions. The remaining samples were prepared using the NEBNext® Enzymatic Methyl-seq (EM-seq) Kit (NEB, Ipswich, MA, USA). EM-seq was performed according to manufacturer's guidelines for standard insert libraries with 14 PCR cycles. Libraries were quantified and quality-checked using the D1000 ScreenTape Analysis Assay (Agilent) before pooling. Paired-end 150 base pair (bp) libraries were pooled in equimolar amounts and sequenced on a NovaSeq6000 (Illumina) (GenomeScan, Leiden). The processing of sequencing data and subsequent analysis of SCNA and cfDNA fragmentation patterns are provided in the Supplemental Methods. Shallow whole-genome sequencing of paired FFPE primary tumor tissue was performed to verify copy number profile concordance and is also described in the Supplemental Methods.

### Statistical analysis
Methylation levels were expressed as $^{2log}$-transformed Cq ratios and presented in violin plots. Tissue methylation levels were compared between two groups using the non-parametric Mann–Whitney *U* test. Methylation levels of each gene in the remaining sample types were compared between healthy controls and patients diagnosed with a benign or malignant ovarian mass using the Kruskal–Wallis test. In the case of a significant Kruskal–Wallis test ($P < 0.05$), this was followed by post-hoc testing of (1) healthy controls versus malignant ovarian masses, and (2) benign versus malignant ovarian masses using the Mann–Whitney *U* test with Bonferroni correction.

The correlation between methylation levels of each DNA methylation marker between paired samples of patients diagnosed with ovarian cancer was assessed using Spearman's rank correlation. Correlation coefficient *r* was defined as very weak ($r = 0.00–0.19$), weak ($r = 0.20–0.39$), moderate ($r = 0.40–0.59$), strong ($r = 0.60–0.79$), or very strong ($r = 0.80–1.00$) and displayed in correlation matrices.

Fragment size profiles were visualized by density plots and analyzed by comparing cfDNA reads of healthy controls and ovarian cancer patients with low (<5%) and high (≥5%) tumor fractions.

Data were collected using Castor EDC and analyzed using R (version 4.0.3 with packages: cowplot, corrplot, dplyr, ggplot, ggpubr, and rstatix). *P* values are two-sided and considered statistically significant when $P < 0.05$. Reported *P* values are Bonferroni corrected when comparing >2 groups (i.e., divided by the number of comparisons). Given the exploratory nature of this study, nine methylation markers were included without correcting for multiple testing.

### Reporting summary
Further information on research design is available in the Nature Portfolio Reporting Summary linked to this article.

## Results
### Study population
A flowchart describing the study overview and sample types used is shown in Fig. 1. Tissue samples of the normal fallopian tube (n = 22) and HGSOC (n = 35) were collected to verify the discriminatory power of qMSP assays. Patient-friendly samples (urines, cervicovaginal self-samples, and clinician-taken cervical scrapes) were prospectively collected from 54 patients undergoing pelvic surgery at a tertiary oncology center because of a highly suspicious ovarian mass. Twenty-nine women were diagnosed with ovarian cancer and 25 with a benign ovarian mass. For comparison, 110 non-paired samples of healthy age-matched controls were collected from different settings, including 30 urines, 40 cervicovaginal self-samples, and 40 clinician-taken cervical scrapes. Clinical characteristics of study participants of which patient-friendly material was collected are summarized in Table 1. From 1 patient, no self-sample was available, and from 8 patients, no cervical scrapes were available. Three samples could not be included in the analysis because of invalid methylation results (*ACTB* Cq ≥ 32; full void urine n = 1, urine supernatant n = 1, self-sample n = 1).

### DNA methylation levels are elevated in cervical scrapes and urine samples of women with ovarian masses
All markers showed clear significant differences when comparing methylation levels in the normal fallopian tube (n = 22) with HGSOC (n = 35) tissues (*P* < 0.0001; Supplemental Fig. 1, Mann–Whitney *U*).

The possibility of ovarian cancer detection in urine by methylation analysis was evaluated by testing nine methylation markers in full void (i.e., unfractionated) urine, urine supernatant, and urine sediment of healthy controls and patients diagnosed with a benign or malignant ovarian mass (Fig. 2 and Supplemental Figs. 2−4). When comparing healthy controls with ovarian cancer patients, three markers showed a significant discrimination in full void urine (*C2CD4D*, *P* = 0.008; *CDO1*, *P* = 0.022; *MAL*, *P* = 0.008, Mann–Whitney *U*), one in urine supernatant (*MAL*, *P* = 0.001) and one in urine sediment (*GHSR*, *P* = 0.018, Mann–Whitney *U*). Benign and malignant masses revealed comparably high methylation levels for most methylation markers, except for *GHSR*. *GHSR* showed significantly elevated methylation levels in the urine sediment of ovarian cancer patients (*P* = 0.024, Mann–Whitney *U*; Fig. 2 and Supplemental Fig. 4).

Similarly, the possibility of ovarian cancer detection in cervicovaginal self-samples and clinician-taken cervical scrapes by methylation analysis was assessed by testing the same methylation markers. While methylation levels of two markers were significantly increased in clinician-taken cervical scrapes of ovarian cancer patients as compared to healthy controls (*C2CD4D*, *P* = 0.001; *CDO1*, *P* = 0.004, Mann–Whitney *U*), benign and malignant ovarian masses could not be distinguished using these markers (Fig. 2 and Supplemental Fig. 5). None of the markers were significantly elevated in cervicovaginal self-samples when comparing these groups (Fig. 2 and Supplemental Fig. 6).

Numbers were insufficient to compare methylation levels between different histological subtypes and stages. Significant differences remained when using a lower *ACTB* Cq threshold of ≤30 for adequate sample quality (Supplemental Figs. 7 and 8). Further studies are required to validate the discriminatory power of the investigated methylation markers as we did not correct for multiple testing of the nine methylation markers in this exploratory phase. Source qMSP methylation data can be found in Supplementary Data 1.

### DNA methylation levels are correlated between paired cervical scrapes and urine samples
DNA methylation levels of genes significantly discriminating between healthy controls and malignant ovarian masses in cervical scrapes and urine (i.e., *C2CD4D, CDO1, GHSR, MAL*) were compared between paired samples to assess their correlation (Supplemental Fig. 9). Paired cervical scrapes and urine were available for 23 ovarian cancer patients. Individual markers in full void urine correlated moderately to strongly with urine supernatant ($r = 0.52–0.61$) and urine sediment ($r = 0.67–0.76$). The full void urine

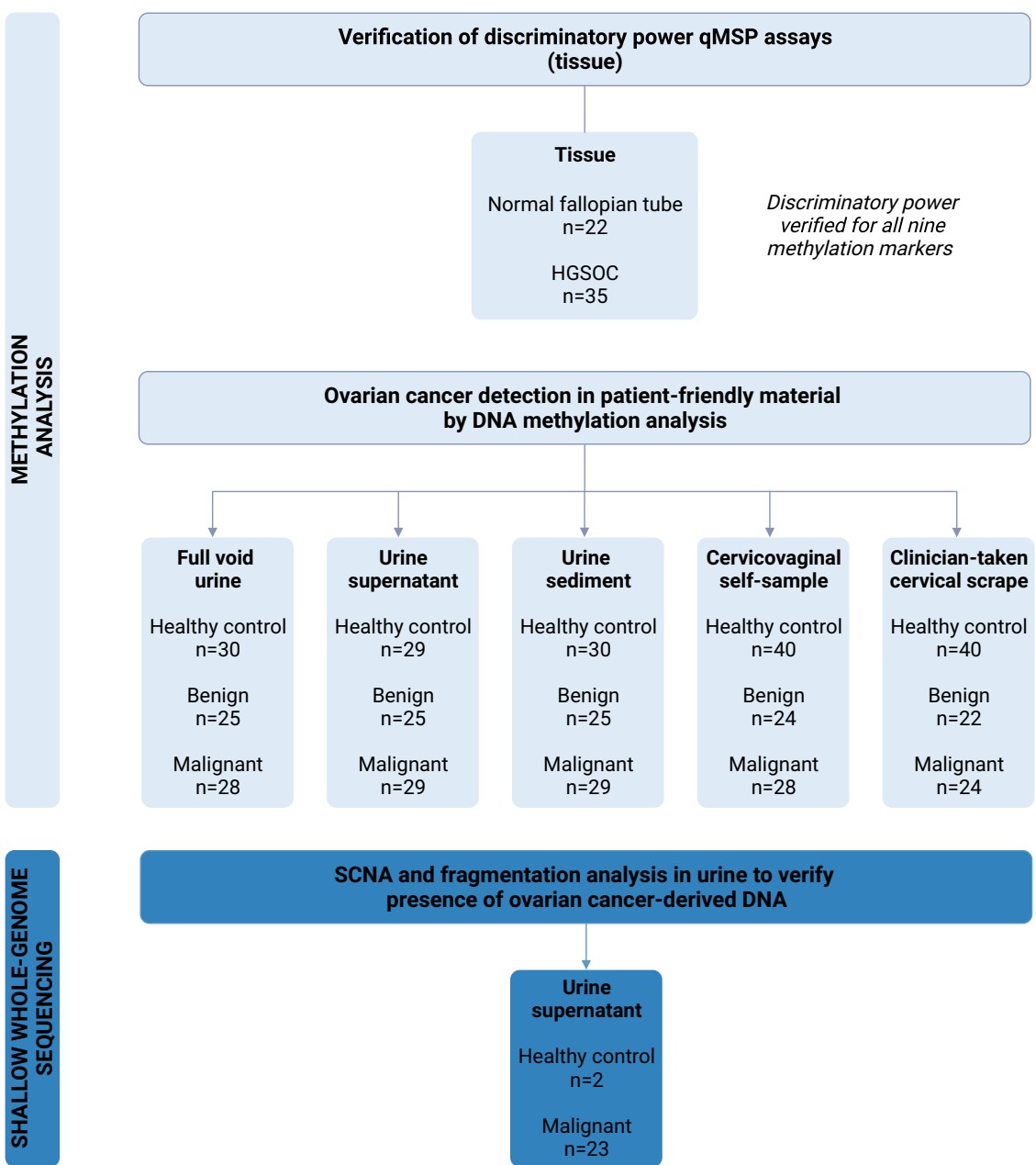

**Fig. 1 | Study flowchart illustrating samples included in the methylation analysis and shallow whole-genome sequencing.** HGSOC high-grade serous ovarian cancer, SCNA somatic copy number aberrations, qMSP quantitative methylation-specific PCR.

showed the best correlation with cervical scrapes ($r = 0.42–0.59$), while a weak correlation was observed between the urine supernatant and cervical scrapes ($r = 0.33–0.45$).

**Copy number aberrations are detectable in urine cell-free DNA**
The presence of ovarian cancer-derived DNA in the urine was verified by analyzing a subset of 25 urine supernatant samples of ovarian cancer patients ($n = 23$) and healthy controls ($n = 2$) by shallow whole-genome sequencing. Sequencing yielded a sufficient read count for all samples (median mapped paired read count of 55,133,492). Shallow whole-genome sequencing coverage and quality statistics per urine sample are provided in Supplementary Data 2. Aberrant genome-wide copy number profiles were found in 4 out of 23 sequenced urine supernatant samples of ovarian cancer patients (Fig. 3 and Supplemental Fig. 10). Copy number profile concordance between urine and the primary tumor tissue was verified for these cases (Supplemental Fig. 10).

The patient with the highest tumor fraction also showed the highest methylation levels of *MAL* in the urine supernatant (Supplemental Fig. 11). In addition, fragment size distributions were analyzed by comparing cfDNA reads of healthy controls and ovarian cancer patients with low and high tumor fractions. Cancer samples with a high tumor fraction ($n = 4$) revealed a shorter modal fragment size of 80 bp as compared to 111 bp in cancer samples with a low tumor fraction ($n = 19$) and controls ($n = 2$; Supplemental Fig. 12).

**Discussion**
Both elevated methylation levels of a subset of markers and SCNA were detected in home-collected urine samples of ovarian cancer patients by targeted qMSP assays and shallow whole-genome sequencing, respectively. Urine is truly non-invasive and unlocks at home collection of liquid biopsy to reduce in-person visits. Yet, an important finding was that methylation levels in benign cases were similarly high, presenting a challenge for the development of clinically useful tests.

**Table 1 | Clinical characteristics of study participants**

| | n | % | Age: median (IQR) |
|---|---|---|---|
| **Ovarian cancer** | 29 | | 59 (56–67) |
| Histology | | | |
| Serous carcinoma | 22 | 75.9% | |
| Low grade | 4 | | |
| High grade | 18 | | |
| Clear cell carcinoma, high grade* | 3 | 10.3% | |
| Carcinosarcoma, high grade | 2 | 6.9% | |
| Endometrioid carcinoma, low grade | 1 | 3.4% | |
| Mucinous carcinoma, low grade | 1 | 3.4% | |
| Stage (FIGO 2014) | | | |
| IIB | 5 | 17.2% | |
| IIC | 1 | 3.4% | |
| IIIA | 5 | 17.2% | |
| IIIB | 4 | 13.8% | |
| IIIC | 12 | 41.4% | |
| IV | 2 | 6.9% | |
| **Benign ovarian mass:** | 25 | | 62 (54–69) |
| Histology | | | |
| Serous cystadeno(fibro)ma | 8 | 32.0% | |
| Mucinous cystadenoma | 6 | 24.0% | |
| Fibroma | 4 | 16.0% | |
| Endometriosis cyst | 4 | 16.0% | |
| Mature teratoma | 3 | 12.0% | |
| **Healthy controls:** | 110 | | |
| Sample type | | | |
| Urine | 30 | | 60 (53–74) |
| Cervicovaginal self-sample | 40 | | 60 (60–60) |
| Clinician-taken cervical scrape | 40 | | 60 (60–60) |

*Including one mixed clear cell and low-grade endometrioid carcinoma.

While we tested for methylation markers described and also by us verified to be associated with ovarian cancer, it was found that when tested in our patient-friendly sample types most of these did not distinguish benign from malignant ovarian masses. Only *GHSR* demonstrated slightly increased methylation levels in the urine sediment. Benign ovarian masses included in this study were highly suspicious for malignancy according to current triage methods (>40% risk of malignancy using the IOTA adnex model) as samples were collected in a tertiary oncology unit. Half of the included patients in our cohort were ultimately diagnosed with a benign ovarian mass, underlining that current triage for referral to tertiary oncology care is suboptimal. The majority of previous studies only included benign controls for methylation marker discovery in tissue but not during marker validation in plasma, as recently reviewed in ref. 15, or benign controls were not age-matched to cancers[26]. Similarly, studies on ovarian cancer detection in cervical scrapes did not include benign controls[16,17]. The inclusion of age-matched patients diagnosed with benign and malignant ovarian masses is essential to accurately assess the clinical value of DNA methylation testing for ovarian cancer detection.

The presence of ovarian cancer-derived DNA in the urine is currently underexplored. So far, only Valle et al. reported on the detection of somatic mutation profiles and *HIST1H2BB/MAGI2* promoter methylation in a small paired series of ascites, blood, tissue, urine, and vaginal swabs of HGSOC patients[34]. Their data on two patients revealed that methylation levels in urinary cfDNA correlated stronger with tissue than with blood, indicating the potential of urine-based ovarian cancer detection.

Unfortunately, the diagnostic potential of ovarian cancer detection in urine could not be determined in the study of Valle et al. as no control samples were included.

In our study, different urine fractions were systematically compared to explore whether a preferred urine sample type for ovarian cancer detection exists. Full void urine most likely contains both genomic and cfDNA, whereas the urine sediment is enriched for genomic DNA and the urine supernatant for transrenally excreted cfDNA[35]. This assumption is confirmed by the strong correlation for *CDO1* between cervical scrapes and urine sediment, while cervical scrapes and urine supernatant correlated weakly to moderately. Most methylation markers significantly differentiated between healthy controls and ovarian cancer patients in the full void urine (3/9), followed by urine supernatant (1/9), and the urine sediment (1/9). These outcomes suggest that tumor-derived methylation signals can originate from genomic DNA as well as transrenally excreted cfDNA. Differences between the urine fractions could potentially be explained by the use of different starting volumes (30 mL for full void urine vs. 15 mL for urine supernatant and sediment). Hence, larger samples sizes and preferably equal starting volumes are needed to determine whether a preferred urine sample type for methylation analysis exists.

Even though altered DNA methylation occurs early during cancer development, detecting methylation signals is challenging due to the limited presence of tumor-derived signals in body fluids, particularly in early-stage cancers. This challenge is highlighted by the recent PATHFINDER and SYMPLIFY studies using methylation-based multi-cancer early detection tests in plasma. The majority of false negatives consisted of early-stage cancers, with early-stage cancer sensitivity ranging from 16.3 to 24.2%[36,37]. In this study, genes with elevated methylation levels in HGSOC tissue, were not always measurable in urine. Our qMSP assays were designed to facilitate the detection of methylation in small DNA fragments present in the urine as shown in our previous studies[8,10,12]. Yet, the current assays may not reach the limit of detection needed for the low tumor-derived methylation signals. Nucleic acids that are released from the bladder epithelium may further dilute the ovarian cancer signal in urine. A higher signal-to-noise ratio could be obtained by targeting a larger panel of methylated regions by methylation sequencing[38]. Alternatively, the sensitivity of PCR-based methylation analysis could be enhanced by using sense-antisense droplet digital PCR or Target Enrichment Long-Probe Quantitative-Amplified Signal format (TELQAS) assays as successfully employed previously for plasma-based ovarian cancer detection[27,39].

Another explanation for the absence of tumor-derived methylation signals of some genes in the urine could be linked to the origin of urinary cfDNA. Urine cfDNA is described to be even shorter as compared to plasma cfDNA (modal size of 82 vs. 167 basepairs)[40]. Differences in fragmentation patterns between plasma and urine are likely caused by Dnase1 cleavage activity in the urine and high concentrations of urea and salt that affect histone-DNA binding[41]. Histone-bound DNA is more protected against degradation as compared to DNA that is not histone-bound[42]. Hypothetically, hypermethylated regions of interest that are not histone-bound could be further degraded and become unmeasurable. We partly accounted for this by including methylation markers with proven diagnostic value in plasma in our selection (i.e., *C2CD4D*[26–28], *CDO1*[27]), which both appeared suitable for ovarian cancer detection in urine.

Clear SCNA profiles harboring common chromosomal gains (e.g., 1q, 3q, 7q, 8q) and losses (e.g., 17p, 19q, 22q) could be obtained from four urine supernatant samples of ovarian cancer patients, verifying the presence of tumor-derived DNA in the urine[43]. Furthermore, a focal amplification at chromosome 19 was identified in the urine of one patient with stage IIIA serous carcinoma, which is a clinically relevant alteration that has previously been described in a subgroup of serous ovarian cancers[44]. Aneuploidy was detected previously in cervical scrape samples of ovarian cancer patients using the PapSEEK test[16].

We also observed shorter fragment sizes in urine supernatant samples with a high tumor fraction, which is another indication for the presence of tumor-derived DNA in the urine, as shown previously in urine samples of

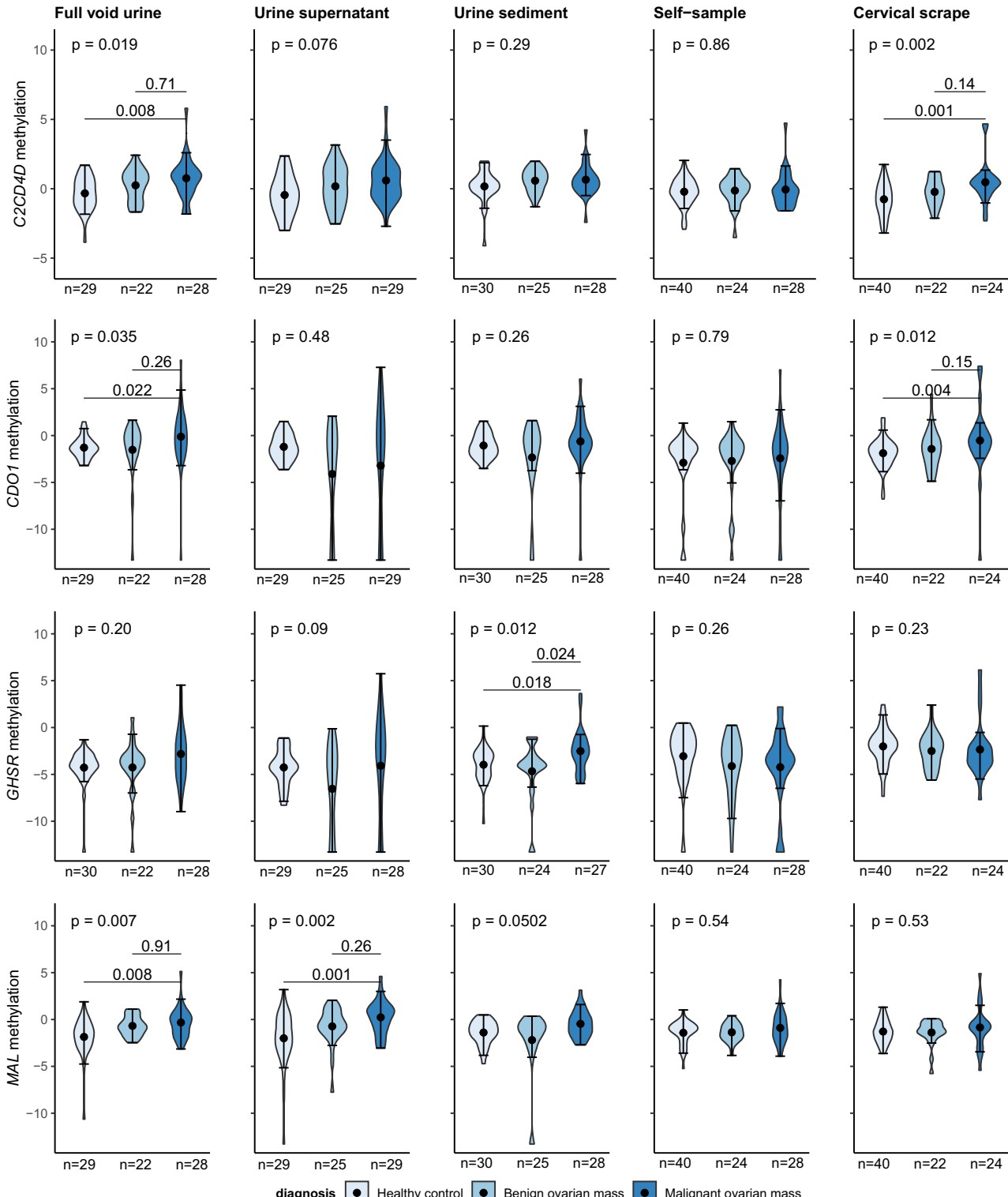

**Fig. 2 | Methylation levels of most discriminating markers *C2CD4D*, *CDO1*, *GHSR*, and *MAL* in full void (unfractionated) urine, urine supernatant, urine sediment, cervicovaginal self-samples, and clinician-taken cervical scrapes of healthy controls and patients diagnosed with a benign or malignant** **ovarian mass.** Methylation levels are expressed by $^{2log}$-transformed Cq ratios. Violin plots represent medians with lower and upper quartile and range whiskers. *P* values shown are Bonferroni corrected (i.e., divided by the number of diagnostic groups compared) and considered statistically significant when <0.05.

glioma patients[40]. While fragment size analysis could potentially enhance ovarian cancer detection in urine, current findings warrant further validation. This study only contained four samples with a tumor fraction >5% and no benign cases were sequenced. Alternatively, leveraging fragmentation patterns for nucleosome footprinting could provide a more robust approach

to distinguish between benign and malignant ovarian masses, as shown previously using plasma cfDNA[45].

Given the feasibility of ovarian cancer detection in cervical scrapes by DNA methylation analysis[14,17], similar findings were expected for self-collected cervicovaginal samples. While *C2CD4D* and *CDO1* distinguished

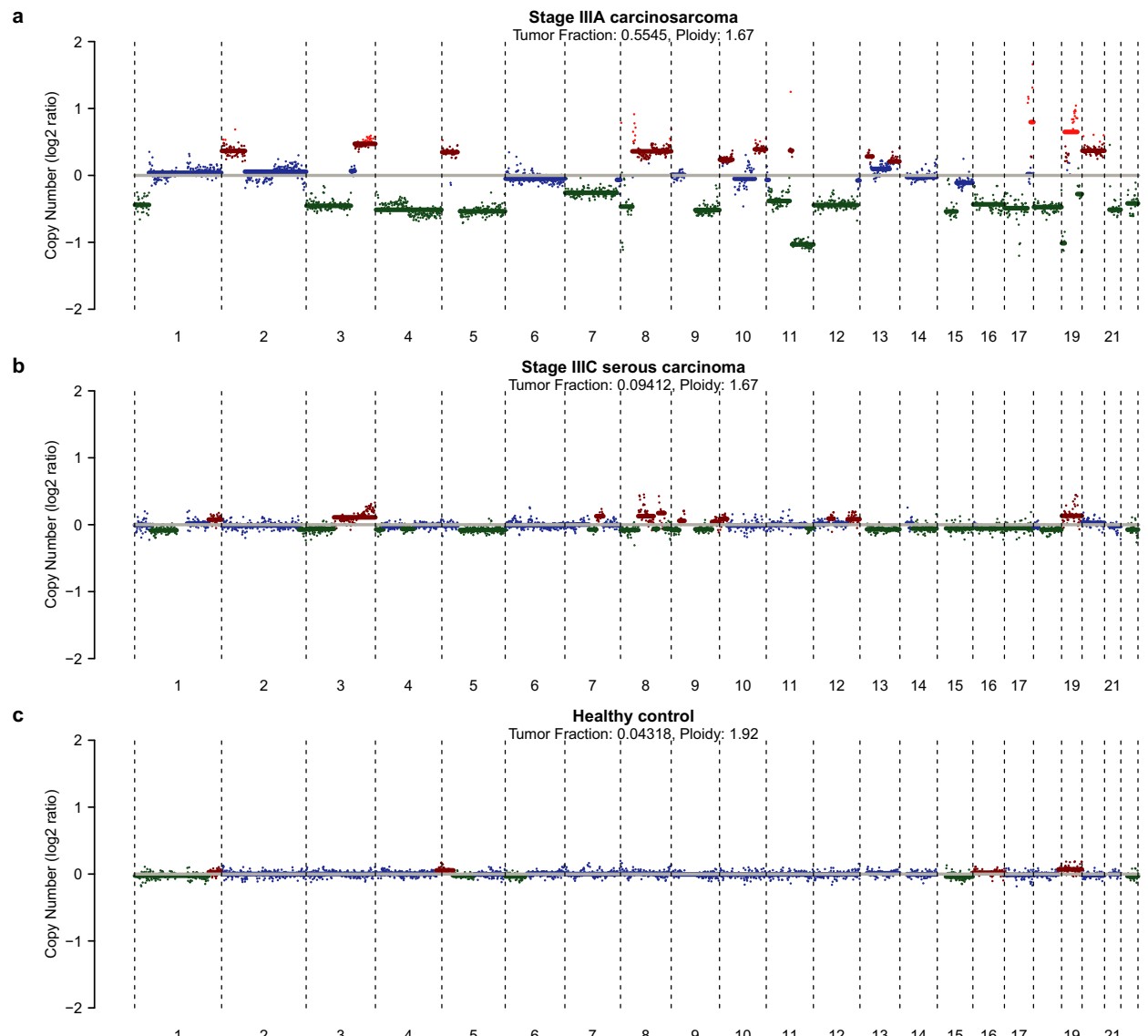

**Fig. 3 | Genome-wide somatic copy number profiles of urine supernatant samples.** Panels show illustrative examples of patients with a stage IIIA carcinosarcoma (**a**), stage IIIC serous carcinoma (**b**), and a healthy control (**c**). Estimated ploidy and tumor fraction are listed at the top of the plot. The *y* axis depicts the log2 tumor to normal ratio.

healthy versus malignant in cervical scrapes, none of the markers showed elevated methylation levels in cervicovaginal self-samples. Our findings are in line with those of van Bommel et al. who reported that mutation analysis in cervicovaginal self-samples of ovarian cancer patients was not feasible[46]. None of the pathogenic mutations found in surgical specimens could be detected in cervicovaginal self-samples. Ovarian cancer signals might be more diluted in cytological specimens collected from areas further away from the ovaries. This was also observed for the PapSEEK test, which detected 45% of ovarian cancers when using intrauterine sampling (Tao brush) as compared to 17% when using endocervical sampling (Pap brush)[16].

Nevertheless, considering our relatively small sample size, we do not exclude the use of cervicovaginal self-samples for ovarian cancer detection yet. The optimization of pre-analytical factors, such as increased input of original sample or improved DNA isolation methods, could enhance the ovarian cancer signal in vaginal samples. Alternatively, a non-tumor DNA driven approach could be useful for ovarian cancer detection in cervicovaginal self-samples, as recently described in ref. 47. Their signature consisted of epigenetic differences in cervical cells and allowed ovarian cancer detection in cervical scrapes with an area under the receiver operating characteristic curve value of 0.76. Larger cohort studies, such as the

Screenwide study[48], will provide further insight into the use of cervicovaginal self-samples for ovarian cancer detection.

Strengths of this study include the collection of a unique paired sample series of both patients diagnosed with a benign ovarian mass and with a malignant ovarian tumor, covering most histological subtypes. Moreover, urine and cervicovaginal self-samples were collected from home to assess the potential of home-based sampling for ovarian cancer. The successful sequencing of urine cfDNA of ovarian cancer patients provides opportunities for future (epi)genome profiling using short- or long-read sequencing technologies. Limitations of the study are the relatively low sample numbers and the lack of early-stage cancers (≤ FIGO stage 2A). Moreover, information on prior benign gynecological disease in healthy control women was not available. Given the heterogeneous nature of benign and malignant ovarian masses, larger sample series and more sophisticated methodologies are needed to conclude on the clinical applicability of home-collected cervicovaginal self-samples and urine to improve the pre-surgical diagnosis of women presenting with an adnexal mass. Third, this study did not have access to paired plasma samples for a direct comparison using DNA-based and other molecular biomarkers (e.g., HE4).

This study supports limited existing data on ovarian cancer detection in cervical scrapes by DNA methylation analysis. Moreover, we are not

**Article**

aware of any previous reports showing that urine yields increased methylation levels of ovarian cancer-associated genes and contains ovarian cancer-derived DNA as demonstrated by SCNA analysis. Our findings support continued research into urine biomarkers for ovarian cancer detection and highlight the importance of including benign ovarian masses in future studies. Molecular biomarker testing in patient-friendly samples could facilitate earlier ovarian cancer detection and triage women presenting with an ovarian mass to manage specialist referral. Yet, further studies investigating alternative urine (methylation) biomarkers are warranted to develop a clinically useful test.

## Data availability

Source qMSP methylation data can be found in Supplementary Data 1. The sequencing dataset generated and analyzed during this study is available in the European Genome-Phenome Archive repository, under accession number EGAD00001010848. Sequencing data is under restricted access to protect study participant anonymity. Access can be requested by contacting the corresponding author. All other data are available from the corresponding author (or other sources, as applicable) on reasonable request.

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

## Acknowledgements

This research was funded by the CCA Foundation and Maarten van der Weijden Foundation, who provided financial support for the conduct of the research and had no role in conducting the research and preparation of the manuscript. RvdH was funded by the Weijerhorst Foundation. NM and FM were funded by the Dutch Cancer Society (grant number KWF: 12822). The authors thank Dr. J.B. Poell, Dr. J.C. Kasius, and Dr. C.H. Mom for the useful discussions and support during funding acquisition. The authors also thank Dr. Y. Kim, J.M.P. Egthuijsen, N. Hogervorst, and N. Evander for technical assistance, Dr. B.I. Lissenberg-Witte for statistical support and A.H. Koch for help during sample collection.

## Author contributions

B.W.: funding acquisition, resources, data curation, formal analysis, investigation, visualization, methodology, writing—original draft. M.S.: Resources, data curation, writing—review and editing. M.B.: conceptualization, supervision, funding acquisition, validation, writing—review and editing, project administration. Y.B.: Investigation, data curation, methodology, writing—review and editing. R.H.: conceptualization, resources, writing—review and editing. C.L.: resources, writing-review and editing. F.D.: resources, writing—review and editing. Y.P.: investigation, writing—review and editing. F.M.: resources, formal analysis, visualization, methodology, writing—review and editing. N.M.: resources, formal analysis, visualization, methodology, writing—review and editing. N.T.: conceptualization, supervision, funding acquisition, resources, validation, writing-review and editing, project administration. R.S.: conceptualization, supervision, funding acquisition, validation, writing—original draft, project administration.

## Competing interests

R.D.M.S. is a minority shareholder of Self-screen BV. The remaining authors declare no competing interests.
