## [Peer Review File · Communications Medicine]

Reviewers' comments:

Reviewer #1 (Remarks to the Author):

There remains no effective early detection test for ovarian cancer. This manuscript attempts to utilise the detection of methylated biomarkers in “patient friendly samples” to distinguish between benign and malignant ovarian masses, and between ovarian cancer and healthy controls. Biospecimens are collected from completed clinical trials and biobanks and methods are robust. The collection of three different biospecimens (urine, cervicovaginal self samples and cervical scrapes) from a limited cohort (n=54) is a strength of the manuscript, as is the inclusion of an age matched benign cohort. This is an important study to publish, despite its largely negative result. The limitations of the study are clearly stated and mostly discussed. Clarification is sought on the following points:

- How is a “patient friendly” sample defined? Can you provide evidence and references that patients, consumers or advocates have identified these methods of biospecimen collection as preferential?
- Some further clarification around the cohorts is needed. The cohort of 54 (25 benign and 29 malignant) is clear, but the 110 unmatched healthy controls are unclear. Are these all from the biobank? Is a prior history of benign gynaecological disease known for these patients? The control cervicovaginal samples are listed as being from high risk HPV negative women – why are they high risk? How is this defined? What are the ages of these patients? Are they pre or post menopausal? Please provide further information on this cohort, split by biospecimen type.
- Is there any possibility of obtaining plasma from the patient enrolled in the clinical trial or the biobank? Comparison with this biospecimen, even in a limited number would provide interesting information on fragmentation and detection of these methylated biomarkers.
- Please rewrite the following confusing sentence - “Patients of which not all three paired sample types (i.e., cervical scrape, cervicovaginal self-sample, and urine) were available were not excluded” – should this be were excluded, eg unless you had all 3 your were out of the study?
- Many of the primer sequences are only “available upon request” - please provide justification for this?
- The different fragmentation in different fluids, and in control urine vs ovarian cancer patient urine is an interesting finding - please add further discussion about the limitation of your methodological approach and opportunities for further refinement.
- Please add further discussion about where you would see ovarian cancer detection via these biospecimens occurring clinically? At a population level? As a gynae-oncology triage step? Please comment on the results of your study and how this may be implemented clinically.

Reviewer #2 (Remarks to the Author):

The authors utilize several patient-friendly collected samples to detect ovarian cancer.

Although assessing DNAm in urine is an interesting concept that has not yet been fully explored, the paper is rather limited by the low number of patients studied and the very limited number of features assessed. In addition, there are several methodological flaws.

In more detail:

Abstract:

In the Methods section, the discovery should be described prior to the validation in patient-friendly samples of women with pelvic masses.

The Results section needs improvement: when the authors refer to healthy women I suspect they mean women with benign pelvic masses – is this correct? The p-values need adjustment for multiple testing (i.e. 9 markers have been tested and hence only $P < 0.006$ would indicate statistical significance – something along these lines is mentioned in line 192, but I am unsure whether this has in fact been implemented).

Introduction:

Using references 4 & 5 the authors refer to trials in general and high-risk populations. Both UKCTOCS and the PLCO trial were largely general population; screening in women at high-risk was assessed in UKFOCSS trial (but in UKFOCSS survival was not an outcome, only stage).

The authors state that “DNA methylation-mediated silencing of tumor suppressor genes occurs early in cancer development and is therefore promising to detect cancer at an early-stage”; the limiting factor so far is not the availability of a cancer-specific markers that are present in early stage disease, but the abundance of such a marker in the system at early stages. It is this low abundance limits the detection of cancers at an early stage – in this context it would also be advisable to quote the recent papers published by Grail in Lancet Oncol (SYMPLIFY) and Lancet (PATHFINDER) indicating the low rates of cancers detected early by utilising cell-free DNA.

Methods:

The authors state “Patients without residual tumor/ovarian mass at time of inclusion ...were excluded”; this implies that women with any type of cancer – even after surgery as long as residual cancer was present – were included in the study?

It is unclear where the materials for the discovery and validation have been collected. The authors indicate that urine and cervicovaginal samples have been obtained from various biobanks; was this for the discovery set? It is essential to describe the source of the materials and the way they had been collected very thoroughly because these aspects are among the biggest sources of bias. There seems to be also a mix of FFPE and frozen tissue.

Supplementary Tables for the discovery and, separately, for the validation sets are required and should provide all relevant details.

The authors utilized a shortened amplicon size for ACTB, which I assume lacks any CpGs, to assess DNA input and the efficacy of the bisulfite modification. Can they provide the length and primers/probe (not sure why this information is excluded in the relevant Supplementary Table?). The authors excluded samples with a ACTB quantification cycle (Cq) ≥ 32 . This is very high. In order to assess reliability/stability of the assay, can the authors also provide the limit of detection and limit of blank for this and all the other reactions (ideally utilising fully methylated DNA diluted in unmethylated DNA, etc). I am unsure how the authors assessed the DNAm status of their positive control (SiHa cell line). Would strongly suggest the use of SssI-treated DNA.

Results:

The number of volunteers included is very low and hence the broad relevance of the finding is rather questionable in particular considering the fact that the cases consisted mostly of patients with advanced stage disease.

Discussion:

It is surprising that the authors did not discuss alternative DNAm detection technologies with a higher signal to noise ratio.

Reviewer #1 (Remarks to the Author):

There remains no effective early detection test for ovarian cancer. This manuscript attempts to utilise the detection of methylated biomarkers in “patient friendly samples” to distinguish between benign and malignant ovarian masses, and between ovarian cancer and healthy controls. Biospecimens are collected from completed clinical trials and biobanks and methods are robust. The collection of three different biospecimens (urine, cervicovaginal self samples and cervical scrapes) from a limited cohort (n=54) is a strength of the manuscript, as is the inclusion of an age matched benign cohort. This is an important study to publish, despite its largely negative result. The limitations of the study are clearly stated and mostly discussed.

We thank the reviewer for the thorough revision of our manuscript and appreciate the given suggestions. We would like to clarify that all samples were collected for the purposes of this pilot study and not within a clinical trial setting. We agree that the use of different diagnostic groups (healthy controls, benign ovarian masses and malignant ovarian masses) in combination with different sample types (urine, self-sample, scrape, tissue) might be confusing to readers. Therefore, a study flowchart has been added to the revised manuscript to clarify the study set-up and included study participants. We also revised the ‘Study population’ results section.

Lines 226-245:

A flowchart describing the study overview and sample types used is shown in Figure 1. Tissue samples of normal fallopian tube (n=22) and HGSOV (n=35) were collected to verify the discriminatory power of qMSP assays. Patient-friendly samples (urines, cervicovaginal self-samples, and clinician-taken cervical scrapes) were prospectively collected from 54 patients undergoing pelvic surgery at a tertiary oncology center because of a highly suspicious ovarian mass. Twenty-nine women were diagnosed with ovarian cancer and 25 with a benign ovarian mass. For comparison, 110 non-paired samples of healthy age-matched controls were collected from different settings, including 30 urines, 40 cervicovaginal self-samples and 40 clinician-taken cervical scrapes. Clinical characteristics of study participants of which patient-friendly material was collected are summarized in Table 1. From 1 patient, no self-sample was available and from 8 patients no cervical scrape was available. Three samples could not be included in the analysis because of invalid methylation results (*ACTB* Cq \geq 32; full void urine n=1, urine supernatant n=1, self-sample n=1).

Figure 1: Study flowchart illustrating included patient-friendly samples included in the methylation analysis and shallow whole-genome sequencing. HGSOC = high-grade serous ovarian cancer; SCNA = somatic copy number aberrations; qMSP = quantitative methylation-specific PCR.

Clarification is sought on the following points:

- How is a “patient friendly” sample defined? Can you provide evidence and references that patients, consumers or advocates have identified these methods of biospecimen collection as preferential?

We agree that a clear definition of a patient-friendly sample type is currently missing and we have added this to the introduction of the revised manuscript. This section now refers to five studies investigating the acceptability and preferences of urine and cervicovaginal self-sampling (Schaafsma et al. 2022, Shin 2019, and Morgan 2019) and the cost-effectiveness of molecular testing in self-collected material (Peremiquel-Trillas et al. 2023, and Huntington 2023).

The following lines have been added to the manuscript:

Lines 88-94:

Urine and cervicovaginal self-samples are defined as patient-friendly sample types as both sampling strategies allow home collection without pain or discomfort. Experiences with these sampling strategies have been evaluated the context of cervical cancer screening, showing that both were considered acceptable and preferred by the majority of women (18-20). Moreover, molecular testing in self-collected material could offer a cost-effective alternative to detect ovarian cancer, as assessed previously for cervical and endometrial cancer diagnostics (21,22).

• Some further clarification around the cohorts is needed. The cohort of 54 (25 benign and 29 malignant) is clear, but the 110 unmatched healthy controls are unclear. Are these all from the biobank? Is a prior history of benign gynaecological disease known for these patients? The control cervicovaginal samples are listed as being from high risk HPV negative women – why are they high risk? How is this defined? What are the ages of these patients? Are they pre or post menopausal? Please provide further information on this cohort, split by biospecimen type.

We thank the reviewer for pointing this out. The methods section describing the age-matched healthy controls has been revised. All healthy control samples were retrieved from biobanks. Healthy control urine samples were obtained from the URIC biobank. URIC participants were selected for eligibility through a questionnaire to exclude controls with a cancer history in the previous 5 years. Furthermore, age, sex, and smoking history were documented. URIC participants were selected on having the same age range as women diagnosed with a benign or malignant ovarian mass. Healthy control cervical scrapes and cervicovaginal self-samples were both obtained from leftover material of the Dutch national cervical cancer screening program. We apologize for the confusion that occurred due to the use of 'high-risk'. All healthy control samples tested negative for high-risk human papillomavirus, where high-risk indicates virus types known to be involved in cervical cancer development (e.g. HPV16 and HPV18). Unfortunately, no information on prior benign disease and menopausal status has been documented for this cohort. Yet, the majority of women were most likely postmenopausal with 93% of healthy control women aged over 50 years.

The following lines have been adjusted.

Lines 128-138:

Control urine samples were obtained from the URIC biobank, including healthy women without any prior cancer diagnosis within the last five years. Control cervicovaginal self-samples and cervical scrapes were retrieved from leftover material of the Dutch national cervical cancer screening program. Healthy control samples were within the same age range as women diagnosed with an ovarian mass and all tested negative for high-risk human papillomavirus (HPV). Information on prior benign gynecological disease and menopausal status was not documented for healthy control women. Yet, the majority of women were most likely postmenopausal with 93% of healthy control women aged over 50 years.

The lack of information on prior gynecological disease in healthy control women has been added to the discussion as a limitation.

Lines 447-449:

Moreover, information on prior benign gynecological disease in healthy control women was not available.

The age of healthy controls is mentioned in Table 1, which describes the clinical characteristics of all study participants. For clarity, this part has been highlighted below.

Table 1: Clinical characteristics of study participants.

	n	%	Age: median (IQR)
Ovarian cancer:	29	(100%)	59 (56 - 67)
Histology			
Serous carcinoma	22	75.9%	
Low-grade	4		
High-grade	18		
Clear cell carcinoma, high-grade*	3	10.3%	
Carcinosarcoma, high-grade	2	6.9%	
Endometrioid carcinoma, low-grade	1	3.4%	
Mucinous carcinoma, low-grade	1	3.4%	
Stage (FIGO 2014)			
IIB	5	17.2%	
IIC	1	3.4%	
IIIA	5	17.2%	
IIIB	4	13.8%	
IIIC	12	41.4%	
IV	2	6.9%	
Benign ovarian mass:	25	(100%)	62 (54 - 69)
Histology			
Serous cystadeno(fibro)ma	8	32.0%	
Mucinous cystadenoma	6	24.0%	
Fibroma	4	16.0%	
Endometriosis cyst	4	16.0%	
Mature teratoma	3	12.0%	
Healthy controls:	110		
Sample type			
Urine	30		60 (53 - 74)
Cervicovaginal self-sample	40		60 (60 - 60)
Clinician-taken cervical scrape	40		60 (60 - 60)

*Including one mixed clear cell and low-grade endometrioid carcinoma.

• *Is there any possibility of obtaining plasma from the patient enrolled in the clinical trial or the biobank? Comparison with this biospecimen, even in a limited number would provide interesting information on fragmentation and detection of these methylated biomarkers.*

We appreciate this suggestion and agree that adding plasma samples to this manuscript would provide interesting information. Unfortunately we have no access to plasma samples from the patients enrolled in this study. The following lines have been adjusted in the discussion section:

Lines 452-454:

This study had no access to paired plasma samples. For future studies, direct comparisons with paired plasma samples using DNA-based and other molecular biomarkers (e.g., HE4) would be informative.

Moreover, we would like to underline that methylation markers *C2CD4D* and *CDO1* have previously shown diagnostic value in plasma (Widschwendter et al. 2017, Marinelli et al. 2022, and Herzog et al. 2023), which is mentioned in lines 395-398: of the discussion section: 'We partly accounted for this by including methylation markers with proven diagnostic value in plasma in our selection (i.e., *C2CD4D*, *CDO1*), which both appeared suitable for ovarian cancer detection in urine.' We added the most recent study of Herzog et al. 2023 in which the diagnostic value of *C2CD4D* in plasma is described to the current manuscript.

• *Please rewrite the following confusing sentence - "Patients of which not all three paired sample types (i.e., cervical scrape, cervicovaginal self-sample, and urine) were available were not excluded" – should this be were excluded, eg unless you had all 3 your were out of the study?*

We agree that this sentence is not clearly formulated. Our main objective in this relatively small cohort was to explore the possibility of ovarian cancer detection in different patient-friendly sample types, including urine, self-samples, and cervical scrapes. We did not aim to directly compare the performance of ovarian cancer detection in these sample types in individual women. Therefore, women of which not all three samples could be collected were **not** excluded. For example, if a self-sample and urine were both collected, but no cervical scrape material was collected, the self-sample and urine were still analyzed in this study.

The following lines of the methods section have been adjusted accordingly:

Lines 123-125:

Patients were included in the study regardless of whether all three paired sample types were available or not. For example, if a cervical scrape was not collected, the urine and self-sample of this patient were still analyzed and included.

Moreover, we added the following lines to the results section to elaborate on the missing samples in the final analysis.

Lines 242-245:

From 1 patient, no self-sample was available, and from 8 patients, no cervical scrape was available. Three samples could not be included in the analysis because of invalid methylation results (*ACTB* Cq ≥ 32 ; full void urine n=1, urine supernatant n=1, self-sample n=1).

• *Many of the primer sequences are only "available upon request" - please provide justification for this?*

Not all primer sequences are listed because of IP reasons. The missing primer sequences can be provided to readers upon reasonable request.

• *The different fragmentation in different fluids, and in control urine vs ovarian cancer patient urine is an interesting finding - please add further discussion about the limitation of your methodological approach and opportunities for further refinement.*

We would like to clarify that only urine supernatant samples were sequenced as this sample type is enriched for cell-free DNA. This has also been added to the flowchart outlining the study in Figure 1.

We agree that fragment size analysis presents an interesting avenue for further research. We elaborated further on opportunities for further research in the revised version of the discussion section.

Lines 409-415:

While fragment size analysis could potentially enhance ovarian cancer detection in urine, current findings warrant further validation. Present study only contained four samples with a tumor fraction $>5\%$ and no benign cases were sequenced. Alternatively, leveraging fragmentation patterns for nucleosome footprinting could provide a more robust approach to distinguish between benign and malignant ovarian masses, as shown previously using plasma cfDNA (45).

• *Please add further discussion about where you would see ovarian cancer detection via these biospecimens occurring clinically? At a population level? As a gynae-oncology triage step? Please comment on the results of your study and how this may be implemented clinically.*

We are happy to read that the early findings of our manuscript encouraged the reviewer to consider potential clinical applications. We kindly refer to the conclusion section of our manuscript where we briefly discuss the potential clinical value of our approach: 'Molecular biomarker testing in patient-friendly samples could facilitate earlier ovarian cancer detection and triage women presenting with an ovarian mass to manage specialist referral.' While our findings support continued research into urine biomarkers for ovarian cancer detection, we intentionally avoided elaborating on the potential clinical applications of our approach given our limited amount of samples. We therefore directed our focus on potential methodological improvements in the discussion section, aiming to develop a clinically useful test.

Yet, if preferred by the reviewer, we are willing to add a more detailed discussion on the potential clinical implementation of ovarian cancer detection in patient-friendly samples. Upon continued development, urine biomarker testing for ovarian cancer could offer a promising approach for early detection and improving pre-surgical diagnosis in women with ovarian masses. Urine biomarker testing could be integrated with current diagnostic modalities, including serum CA125 testing and transvaginal ultrasonography. This approach would be particularly valuable for high-risk women, including women presenting with an ovarian mass and *BRCA1/2* mutation carriers.

Reviewer #2 (Remarks to the Author):

The authors utilize several patient-friendly collected samples to detect ovarian cancer. Although assessing DNAm in urine is an interesting concept that has not yet been fully explored, the paper is rather limited by the low number of patients studied and the very limited number of features assessed. In addition, there are several methodological flaws.

In more detail:

Abstract:

In the Methods section, the discovery should be described prior to the validation in patient-friendly samples of women with pelvic masses.

We appreciate that the reviewer has assessed our manuscript in great detail.

We would kindly like to note that this exploratory study does not include a discovery and validation cohort. Methylation markers were selected from previous work on cervical, endometrial and ovarian cancer detection in patient-friendly sample types, as described in the introduction: 'Methylation markers considered suitable for the detection of ovarian cancer included a combination of markers described in studies on cervical and endometrial cancer detection in patient-friendly sample types (*GALR1*, *GHSR*, *MAL*, *PRDM14*, *SST*, and *ZIC1*^{10,23-25}), and ovarian cancer detection in cervical scrapes and plasma (*C2CD4D*, *CDO1*, *NRN1*^{17,26-28}).'

We first verified our qMSPs developed for those 9 markers in tissue specimens to determine their discriminatory power. Next, we assessed the possibility of ovarian cancer detection by DNA methylation analysis of these 9 markers in different urine fractions, self-collected cervicovaginal samples and cervical scrapes. Lastly, we verified the presence of ovarian cancer-derived DNA in the urine by shallow whole-genome sequencing for copy number profiling and fragment size analysis. For clarification purposes, a study flowchart has been added to the revised version of the manuscript (Figure 1).

Figure 1: Study flowchart illustrating included patient-friendly samples included in the methylation analysis and shallow whole-genome sequencing. HGSOC = high-grade serous ovarian cancer; SCNA = somatic copy number aberrations; qMSP = quantitative methylation-specific PCR.

The Results section needs improvement: when the authors refer to healthy women I suspect they mean women with benign pelvic masses – is this correct?

We thank the reviewer for raising this question and apologize for any confusion. When referring to healthy women, we refer to the non-paired material collected from age-matched healthy control women retrieved from the URIC biobank (urine) and leftover material of the Dutch national cervical cancer screening program (self-samples and scrapes). Women diagnosed with benign pelvic masses are referred to as 'benign ovarian masses'.

In two instances, only 'controls' was stated, instead of 'healthy controls'. The results section has been revised accordingly for clarification:

Lines 273-277:

While methylation levels of two markers were significantly increased in clinician-taken cervical scrapes of ovarian cancer patients as compared to healthy controls (*C2CD4D*, $p=0.001$; *CDO1*, $p=0.004$, Mann-Whitney U), benign and malignant ovarian masses could not be distinguished using these markers (Figure 1, Supplemental Figure 5).

Lines 295-298:

DNA methylation levels of genes significantly discriminating between healthy controls and malignant ovarian masses in cervical scrapes and urine (*i.e.*, *C2CD4D*, *CDO1*, *GHSR*, *MAL*) were compared between paired samples to assess their correlation (Supplemental Figure 7).

Moreover, the legend of Figure 1 (Figure 2 in revised manuscript) has been adjusted to define the different diagnostic groups more clearly.

diagnosis  Healthy control  Benign ovarian mass  Malignant ovarian mass

The p-values need adjustment for multiple testing (i.e. 9 markers have been tested and hence only $P < 0.006$ would indicate statistical significance – something along these lines is mentioned in line 192, but I am unsure whether this has in fact been implemented).

We thank the reviewer for the critical comment on our statistical analyses. We sought the expertise of a senior biostatistician in our center (dr. Birgit I. Lissenberg-Witte) to discuss the accuracy of our statistical analyses.

There is currently no consensus on when correcting for multiple testing is required. Whether correcting for the testing of nine markers is necessary depends on several factors, of which the study set-up is most important. In explorative studies, like ours, it is generally accepted to not correct for multiple testing when various markers are tested to make sure no potentially significant markers are overlooked. In validation studies it is common to be more strict, as these studies focus on testing previously defined hypotheses.

We added the following lines to methods section to be transparent about this:

Lines 221-222:

Given the exploratory nature of this study, nine methylation markers were included without correcting for multiple testing.

Additionally, the following lines were added to the results section of the manuscript to ensure methylation testing results are interpreted with caution:

Lines 281-283:

Further studies are required to validate the discriminatory power of the investigated methylation markers as we did not correct for multiple testing of the nine methylation markers in this exploratory phase.

We would like to clarify to which p -value correction we refer in lines 203-208: 'Methylation levels of each gene in the remaining sample types were compared between healthy controls and patients diagnosed with a benign or malignant ovarian mass using the Kruskal-Wallis test. In case of a significant Kruskal-Wallis test ($p < 0.05$), this was followed by post-hoc testing of 1) healthy controls versus malignant ovarian masses, and 2) benign versus malignant ovarian masses using the Mann-Whitney U test with Bonferroni correction.'

For each marker and each sample type, a Kruskal-Wallis test was performed to assess whether a significant difference exists between the three groups overall (i.e. between healthy controls, benign ovarian masses and malignant ovarian masses). In case of a significant p -value, two post-hoc tests were performed to examine which groups significantly differ from each other (i.e. healthy control vs. malignant ovarian mass, and benign vs. malignant ovarian mass). The p -values obtained were then corrected using a Bonferroni correction, in which the p -values are divided by the number of comparisons (2).

An example of R code and the corresponding graph are provided below:

```
stat.test <- OC_Scrape %>%  
  wilcox_test(C2CD4D.log ~ diagnosis, ref.group = "Malignant") %>%  
  adjust_pvalue(method = "bonferroni") %>%  
  mutate(y.position = c(4,7))
```

We added the following sentences to the methods section to clarify how p -values are reported:

Lines 219-222:

Reported p -values are Bonferroni corrected when comparing >2 groups (*i.e.* divided by the number of comparisons).

We also adjusted the legend of Figure 2 by adding the following:

Lines 289-291:

P -values shown are Bonferroni corrected (*i.e.* divided by the number of diagnostic groups compared) and considered statistically significant when <0.05.

Introduction:

Using references 4 & 5 the authors refer to trials in general and high-risk populations. Both UKCTOCS and the PLCO trial were largely general population; screening in women at high-risk was assessed in UKFOCSS trial (but in UKFOCSS survival was not an outcome, only stage).

We thank the reviewer for pointing this out. The introduction section has been adapted accordingly:

Lines 70-73:

Although more ovarian cancer patients were diagnosed at an earlier stage with screening strategies using conventional imaging and/or serum biomarkers (*e.g.*, CA-125), this did not translate into reduced overall cancer-specific mortality in general populations^{4,5}.

The authors state that “DNA methylation-mediated silencing of tumor suppressor genes occurs early in cancer development and is therefore promising to detect cancer at an early-stage”; the limiting factor so far is not the availability of a cancer-specific markers that are present in early stage disease, but the abundance of such a marker in the system at early stages. It is this low abundance limits the detection of cancers at an early stage – in this context it would also be advisable to quote the recent papers published by Grail in Lancet Oncol (SYMPLIFY) and Lancet (PATHFINDER) indicating the low rates of cancers detected early by utilising cell-free DNA.

The reviewer raises a critical challenge in liquid biopsy which is currently missing. We agree that the detection of early-stage cancers is largely limited by the low amounts of tumor-derived cfDNA in body fluids. We believe that describing this limiting factor suits the discussion section in which we elaborate on challenges of methylation detection in urine. We thank the reviewer for providing references that strengthen this message and added both to the revised version of the manuscript.

Lines 369-375:

Even though altered DNA methylation occurs early during cancer development, detecting methylation signals is challenging due to the limited presence of tumor-derived signals in body fluids, particularly in early-stage cancers. This challenge is highlighted by the recent PATHFINDER and SYMPLIFY studies using methylation-based multi cancer early detection tests in plasma. The majority of false negatives consisted of early-stage cancers, with early stage cancer sensitivity ranging from 16.3 to 24.2% (36, 37).

Methods:

The authors state “Patients without residual tumor/ovarian mass at time of inclusion ...were excluded”; this implies that women with any type of cancer – even after surgery as long as residual cancer was present – were included in the study?

We apologize for this confusion and removed this sentence from the revised version of the manuscript. This sentence now reads as follows:

Lines 118-120:

Patients who could not collect cytological or urine samples prior to surgery were excluded from participation.

All samples were collected from patients with post-operatively confirmed ovarian cancer or a benign ovarian mass, as stated in lines 113-116: Samples were collected from patients who underwent pelvic surgery with post-operatively confirmed ovarian cancer of any stage and histological subtype, and patients with a benign ovarian mass who were referred to a highly specialized tertiary oncology unit for further assessment.’ Also, all samples were collected prior to surgery, as stated in lines 116-118: ‘Patients scheduled for pelvic surgery, involving exploratory laparotomy to determine the origin of their ovarian mass or cytoreductive surgery, were asked to collect samples prior to surgery.’

It is unclear where the materials for the discovery and validation have been collected. The authors indicate that urine and cervicovaginal samples have been obtained from various biobanks; was this for the discovery set? It is essential to describe the source of the materials and the way they had been collected very thoroughly because these aspects are among the biggest sources of bias.

As mentioned in our previous response, this study does not include a discovery and validation cohort. A study flowchart has been added to the revised version of the manuscript to clarify this.

Urine, cervicovaginal self-samples and clinician-collected cervical scrapes were prospectively collected within the SOLUTION1 study from patients with a highly suspicious ovarian mass, as described in lines 108-110. Urine samples of age-matched healthy control women were retrieved from the URIC biobank, as stated in lines 128-129. Cervicovaginal self-samples and cervical scrapes of healthy control women were retrieved from leftover material from the Dutch national cervical cancer screening program, as described in lines 129-132. A thorough description of the sample collection and processing prior to downstream analysis is given in the Supplementary Methods. We also provide this text below in case the reviewer had no access to the Supplementary Methods file:

Sample collection and processing

Urine and cervicovaginal self-samples were collected at home for which all participants received a package including materials needed for collection and transport. Participants were instructed to collect urine before the cervicovaginal self-sample. Cervicovaginal self-samples were collected according to the provided user manual using the Evalyn® brush (Rovers Medical Devices, Oss, The Netherlands), which is a clinically validated self-sampling method¹. Urine was collected in 3x30 mL tubes containing the storage buffer Ethylenediaminetetraacetic acid (EDTA; final concentration 40mM) to preserve nucleic acids during transport. Clinician-taken cervical scrapes were collected prior to surgery using a Cervex-Brush (Rovers Medical Devices) and directly placed in 10 mL Thinprep PreservCyt medium (Hologic, Marlborough, MA, US). Samples were sent to the Pathology department of Amsterdam UMC, location VUmc, within 72 hours by regular mail and processed directly after arrival.

Urine was processed as described in our previously validated processing and storage protocol². Briefly, a total of 15 mL of urine was centrifuged at 3000g for 10 min to separate the urine into two fractions: the urine supernatant and urine sediment. Both fractions and the remaining full void (*i.e.* unfractionated) urine were stored at -20°C. Cytological samples were processed as described previously for cervical³ and endometrial cancer⁴. Cervicovaginal self-samples were stored in 1.5 mL ThinPrep PreservCyt medium upon arrival. Cervicovaginal self-samples and cervical scrapes were stored at 4°C.

Formalin-fixed paraffin-embedded (FFPE) and fresh frozen tissue specimens were consecutively sectioned of which the first and last sections were Hematoxylin and Eosin (H&E)

stained for histopathological review by a pathologist to confirm the presence of ovarian cancer or normal fallopian tube tissue.

The processing of urine, cervicovaginal self-samples and clinician-taken cervical scrapes has also been described previously in our work on cervical (pre)cancer (van den Helder 2022 Clinical Cancer Res.) and endometrial cancer (Wever 2023 International Journal of Cancer) detection in patient-friendly material, as referred to in lines 146-148: 'The sample collection, processing, DNA extraction, and bisulfite modification procedures were carried out as described previously for cervical (8,32) and endometrial cancer (10,24).'

There seems to be also a mix of FFPE and frozen tissue.

The reviewer correctly points out that a mix of FFPE and fresh frozen tissue specimens were used. We would like to inform the reviewer that only the HGSOE tissue series consisted of a mixture of FFPE and fresh frozen material. Before combining both types of tissue material, we confirmed that both types of material showed largely similar outcomes for methylation analysis when comparing FFPE (n=12) and fresh frozen material (n=23), except for *C2CD4D* ($p = 0.013$) and *MAL* ($p = 0.014$).

Tissue material (FFPE vs. fresh frozen)

Additionally, we confirmed that a statistically significant difference between normal fallopian tube (n=22) and HGSOE (n=12) was still observed when only FFPE tissue material was included.

If preferred by the reviewer, we are willing to substitute Supplementary Figure 1 by the figure provided below to eliminate the mix of FFPE and fresh frozen tissue material of HGSOCs.

Supplementary Tables for the discovery and, separately, for the validation sets are required and should provide all relevant details.

We thank the reviewer for this suggestion. All available and relevant documented clinical characteristics of the study participants of which patient-friendly material was collected are provided in Table 1. As mentioned in the response to Reviewer #1, information on potential prior benign gynecological disease and menopausal status of healthy control women was not available. Yet, the majority of women were most likely postmenopausal with 93% of healthy control women aged over 50 years, which has now been added to the methods section:

Lines 135-138:

Information on prior benign gynecological disease and menopausal status was not documented for healthy control women. Yet, the majority of women were most likely postmenopausal with 93% of healthy control women aged over 50 years.

Missing information on prior benign gynecological disease has been added as limitation to the discussion section:

Lines 447-449:

Moreover, information on prior benign gynecological disease in healthy control women was not available.

The authors utilized a shortened amplicon size for ACTB, which I assume lacks any CpGs, to assess DNA input and the efficacy of the bisulfite modification. Can they provide the length and primers/probe (not sure why this information is excluded in the relevant Supplementary Table?).

Both ACTB designs are not shown because of IP reasons, but are available upon reasonable request. We confirm that the ACTB primers and probe lack CpG sites to adequately correct for DNA input and ensure modification efficacy. All ACTB sequences are designed within the same genomic region and the same probe is used in all multiplexes. The length of ACTB primers and probe in both multiplexes are provided in the table below.

Multiplex	Target	Length fw primer	Length rev primer	Length probe
1,2	ACTB	23	21	19
3	ACTB	20	19	19

The authors excluded samples with a ACTB quantification cycle (Cq) ≥ 32 . This is very high. In order to assess reliability/stability of the assay, can the authors also provide the limit of detection and limit of blank for this and all the other reactions (ideally utilising fully methylated DNA diluted in unmethylated DNA, etc). I am unsure how the authors assessed the DNAm status of their positive control (SiHa cell line). Would strongly suggest the use of Sssl-treated DNA.

We thank the reviewer for being critical about the reliability and stability of our multiplex qMSP assays. We would like to note that no consensus exists on which threshold should be used for excluding samples and that this threshold may vary between different studies. For example, in the study of Wu et al. 2019, exploring ovarian cancer detection in cervical scrapes by qMSP, an even higher threshold for ensuring sufficient sample input of Cq 36 was used. The threshold of ACTB ≥ 32 for sample exclusion is based on our previous studies on bladder, cervical, colorectal, endometrial, and lung cancer detection in patient-friendly material, including urine. Moreover, we would also like to inform the reviewer that the majority of ACTB Cq values were around 25-27, as shown below by the median (IQR) ACTB Cq per multiplex per sample type.

Sample type	ACTB multiplex 1	ACTB multiplex 2	ACTB multiplex 3
Full void urine	25.46 (1.05)	25.31 (1.27)	25.07 (1.3)
Urine supernatant	27.26 (3.1)	26.76 (2.83)	27.07 (3.06)
Urine sediment	25.66 (1.37)	25.4 (1.92)	24.95 (1.41)
Self-sample	26.35 (1.36)	25.31 (1.27)	25.45 (1.04)
Scrape	25.64 (1.1)	24.83 (0.94)	24.82 (1.12)

Only runs in which our negative control H2O shows no signal are accepted (undetermined; ≥ 45) for each target. We assessed the reproducibility and PCR efficiency of our assays from 20 to 0.1 ng methylated SiHa DNA, of which the results are presented in Supplementary Table 2 (also shown below).

Supplementary Table 2: Analytical validation of multiplex quantitative methylation-specific PCR assays.

Multiplex	Target	Slope	R2	Efficiency (%)
1	GHSR	-3,38	1,00	97,52
1	SST	-3,24	0,99	103,69
1	ZIC1	-3,23	0,99	104,00
1	ACTB	-3,39	0,99	97,26
2	CDO1	-3,21	0,99	104,78
2	MAL	-3,28	0,98	101,89
2	PRDM14	-3,37	0,99	98,15
2	ACTB	-3,38	0,99	102,39
3	C2CD4D	-3,46	0,98	94,56
3	GALR1	-3,27	0,99	102,03
3	NRN1	-3,36	0,99	98,64
3	ACTB	-3,38	0,99	97,76

Data is based on serial dilution series of bisulfite treated methylated DNA from the SiHa cell line (100, 50, 10, 5, 1, 0.5%) within the range of 20 to 0.1 ng.

Methylation positivity of the SiHa cell line for the methylation markers included in this study was verified using qMSP. The Ct values of 100% SiHa per target have been provided below.

Multiplex	Target	Ct 100% SiHa
1	GHSR	23,96
1	SST	26,23
1	ZIC1	23,57
1	ACTB	24,93
2	CDO1	25,05
2	MAL	25,90
2	PRDM14	27,77
2	ACTB	25,82
3	C2CD4D	30,51
3	GALR1	24,13
3	NRN1	26,62
3	ACTB	25,54

Since *C2CD4D* shows a relatively high Ct value for SiHa, we provide extra information to the reviewer to verify the reliability and stability of this assay. Underneath, we added amplification plots of *C2CD4D* using a dilution series of double stranded gBlock gene fragments, containing the target amplicon from 0.4 to 10⁵ copies, showing 101.5% PCR efficiency at a slope of -3.3 and R2 of 0.99. These double stranded gBlock gene fragments were also used as positive control in each qMSP run, serving the same purpose as Sssl-treated DNA.

Moreover, we provide a dilution series of bisulfite treated Caov-3 DNA (ovarian cancer cell line) for this target, showing 92.4% PCR efficiency, at a slope of -3.5 and R2 of 0.99.

Amplification Plot

Standard Curve

Target: C2CD4D Slope: -3.52 Y-Inter: 35.148 r^2 : 0.992 Eff%: 92.364 Error: 0.161

Results:

The number of volunteers included is very low and hence the broad relevance of the finding is rather questionable in particular considering the fact that the cases consisted mostly of patients with advanced stage disease.

We agree that the number of included study participants is relatively low and largely consists of women with advanced stage disease, which has been addressed in the limitations section of the discussion in lines 445-447: 'Although we have demonstrated the potential diagnostic value of urine for ovarian cancer, this study is limited by still relatively low sample numbers and the lack of early-stage cancers (\leq FIGO stage 2A).'

Despite relatively low sample numbers, we would like to remark that, to the best of our knowledge, this exploratory study is the first to describe copy number profiling in urine samples of women diagnosed with ovarian cancer. Moreover, only one previous study has described methylation analysis in urine of ovarian cancer patients on two urine samples without including controls (Valle et al. 2020).

Discussion:

It is surprising that the authors did not discuss alternative DNAm detection technologies with a higher signal to noise ratio.

We thank the reviewer for this valuable suggestion. Alternative technologies to assess DNA methylation with a higher signal-to-noise-ratio have been added to the revised version of the discussion section.

Lines 381-386:

A higher signal-to-noise ratio could be obtained by targeting a larger panel of methylated regions by methylation sequencing (38). Alternatively, sensitivity of PCR-based methylation analysis could be enhanced by using sense-antisense droplet digital PCR or Target Enrichment Long-Probe Quantitative-Amplified Signal format (TELQAS) assays as successfully employed previously for plasma-based ovarian cancer detection (27, 39).

Reviewers' comments:

Reviewer #1 (Remarks to the Author):

I thank the authors for comprehensively addressing all of my concerns and updating the manuscript appropriately.

Reviewer #2 (Remarks to the Author):

The authors have provided a detailed response to my comments.

Nevertheless I am still uncertain how useful the manuscript is in terms of deciding whether any of the patient-friendly samples should be further investigated - currently the most important unmet need is to discriminate patients with a malignant adnexal tumour from those with a benign pelvic mass and none of their markers seem to achieve this; the question is whether this is due to a true lack of specific signal or due to a lack of appropriate methodology (i.e. DNAm test methods or sample collection/storage).

It is also difficult to understand why the primers/probes cannot be revealed (i.e. ACTB is only a reference gene and has been utilised in the past by several groups).

It would also be important to provide data for ACTB < ct28 (or <ct30) samples only - this would allow to judge whether the few significant results remain significant in the presence of adequate input DNA. The authors should also comment on whether ACTB ct values differ between the different groups and whether ACTB levels correlate with storage duration/conditions of the samples. It is rather unusual to utilise different ACTB reactions for different multiplex reactions.

Reviewer #2 (Remarks to the Author):

Comment 2.1

The authors have provided a detailed response to my comments. Nevertheless I am still uncertain how useful the manuscript is in terms of deciding whether any of the patient-friendly samples should be further investigated - currently the most important unmet need is to discriminate patients with a malignant adnexal tumour from those with a benign pelvic mass and none of their markers seem to achieve this; the question is whether this is due to a true lack of specific signal or due to a lack of appropriate methodology (i.e. DNAm test methods or sample collection/storage).

Response 2.1

We thank the reviewer for the careful review of our revised manuscript. We agree with this statement and acknowledge that current data do not support the use of methylation-based tests in patient-friendly material to discriminate benign and malignant ovarian masses. Yet, although sample numbers are low, the presence of ovarian cancer-derived DNA in urine was demonstrated by the analysis of copy number aberrations. Therefore, we believe that it is worthwhile to further explore the clinical utility of patient-friendly samples for this purpose using more sophisticated methodologies and larger sample series. The analysis of urine samples of women diagnosed with a benign and malignant ovarian mass using sequencing-based methods are required to fully explore the potential of urine for this clinical need.

To better clarify the limitations of our study we have modified the following lines in the discussion:

Lines 434-443:

Limitations of the study are the relatively low sample numbers and the lack of early-stage cancers (\leq FIGO stage 2A). Moreover, information on prior benign gynecological disease in healthy control women was not available. Given the heterogeneous nature of benign and malignant ovarian masses, larger sample series and more sophisticated methodologies are needed to conclude on the clinical applicability of home-collected cervicovaginal self-samples and urine to improve the pre-surgical diagnosis of women presenting with an adnexal mass. Thirdly, this study did not have access to paired plasma samples for a direct comparison using DNA-based and other molecular biomarkers (e.g., HE4).

Comment 2.2

It is also difficult to understand why the primers/probes cannot be revealed (i.e. ACTB is only a reference gene and has been utilised in the past by several groups).

Response 2.2

Although we would prefer to be transparent about all reactions, this is restricted by a pending patent application including not only the ACTB reaction, but also the GALR1 and NRN1 reactions. Once this invention is released, primer/probe sequences will be available to the scientific community. Before that time, primer/probe sequences are available upon reasonable request following a material transfer agreement. The same holds true for GHSR and ZIC1. For your confirmation, we would like to inform that sequences have previously been provided to researchers when we were approached for this (e.g. DOI: [10.1186/s13148-023-01517-6](https://doi.org/10.1186/s13148-023-01517-6))

Comment 2.3

It would also be important to provide data for ACTB < ct28 (or <ct30) samples only - this would allow to judge whether the few significant results remain significant in the presence of adequate input DNA. The authors should also comment on whether ACTB ct values differ between the different groups and whether ACTB levels correlate with storage duration/conditions of the samples.

Response 2.3

We thank the reviewer for the critical review of the ACTB threshold used. We would kindly like to share the ACTB Ct levels per multiplex and visualized this per sample type and diagnostic category.

The boxplot provided below shows virtually equal ACTB Ct values between the different diagnostic categories for full void urine, self-samples, and scrapes. While a slight increase is seen in urine sediment samples, this is more pronounced for urine supernatant samples of women with a benign or malignant ovarian mass. The potential explanation for higher ACTB Ct levels in urine supernatant is twofold: 1) the nature of the sample type, and 2) the use of different amounts of starting volume.

1. While other sample types contain mostly genomic DNA (urine sediment, self-sample and scrape) or cell-free and genomic DNA (full void urine), urine supernatant is enriched for cfDNA only. Therefore, the total DNA yield is generally lower when extracting DNA from urine supernatant samples. The ACTB reference gene is used to ensure adequate amounts of input DNA.
2. The use of different urine supernatant input amounts during DNA isolation could explain the increased ACTB Ct values in women diagnosed with a benign or malignant ovarian mass. For controls, a total input of 40 mL was available and used for DNA isolation. However, for women diagnosed with a benign or malignant ovarian mass, only 15 mL urine supernatant was available for DNA isolation. Therefore, the total DNA yield differs between these groups and less DNA was available for women diagnosed with a benign or malignant ovarian mass. ACTB is used as internal control to compute Ct ratio's, where the methylation marker level is calculated following the comparative Ct method. This corrects for different amounts of starting material and input DNA used. As indicated in our previous rebuttal, accurate quantification of targets and ACTB was verified before analyzing clinical samples with qMSP efficiencies ranging between 97 and 104% (Supplemental Table 2).

If preferred by the reviewer, we are willing to add this figure to the revised version of the manuscript as Supplementary Figure 11.

We continued with the ACTB Ct cutoff of 32 based on previous work. In a similar study, we utilized urine supernatant samples of patients diagnosed with non-metastatic non-small cell lung cancer (Detection of non-metastatic non-small-cell lung cancer in urine by methylation-specific PCR analysis: A feasibility study - ScienceDirect). Here, the tumor signal also originates from transrenally excreted material. We therefore opted for handling these data with the same threshold. While we prefer to use a ACTB Ct threshold of Ct 32, we are happy to provide the reviewer data when a threshold of Ct 30 is used (indicated by the green line in the figure above).

Please find a revised version of Figure 2 below when a ACTB threshold of Ct 30 was used, showing that the few significant results remain significant. If preferred by the reviewer, we are willing to add this figure to the revised version of the manuscript as Supplementary Figure 12.

Lastly, we would like to add that the same collection and storage methods were used for all diagnostic categories. Scrapes were directly placed in thinprep after collection. Self-samples were collected from home and also placed in thinprep after arrival in the laboratory. Urine samples were collected from home using EDTA as a preservative and directly processed after arrival. The differences between the urine fractions could be explained by the use of different starting volumes (30 mL for full void urine vs. 15 mL for urine supernatant and urine sediment). Potentially, higher input volumes (i.e. > 15 mL) are needed to detect ovarian-cancer associated signals. This note has been added to the revised version of the manuscript.

Lines 352-356:

Differences between the urine fractions could potentially be explained by the use of different starting volumes (30 mL for full void urine vs. 15 mL for urine supernatant and sediment). Hence, larger samples sizes and preferably equal starting volumes are needed to determine whether a preferred urine sample type for methylation analysis exists.

Comment 2.5

It is rather unusual to utilise different ACTB reactions for different multiplex reactions.

Response 2.5

We agree that the use of similar ACTB reactions is preferred. We would like to let the reviewer know that both ACTB reactions are designed within the same genomic region. Our lab is continuously evolving and moving toward optimal qMSP designs for the amplification of highly fragmented urine DNA. The qMSP designs with short amplicon sizes are preferred to facilitate the detection of highly fragmented urine DNA. Therefore, during the course of this study, a novel ACTB design was developed with a shortened amplicon size of 68 base pairs.

Yet, we would like to underline that ACTB Ct values were highly similar amongst the different reactions. This is illustrated by the median Ct value (IQR) per multiplex, per sample type. Multiplex 1 and 2 utilize the same ACTB reaction (amplicon size 68 base pairs), while multiplex 3 incorporated a larger design (amplicon size 108 base pairs).

Sample type	ACTB multiplex 1	ACTB multiplex 2	ACTB multiplex 3
Full void urine	25.46 (1.05)	25.31 (1.27)	25.07 (1.3)
Urine supernatant	27.26 (3.1)	26.76 (2.83)	27.07 (3.06)
Urine sediment	25.66 (1.37)	25.4 (1.92)	24.95 (1.41)
Self-sample	26.35 (1.36)	25.31 (1.27)	25.45 (1.04)
Scrape	25.64 (1.1)	24.83 (0.94)	24.82 (1.12)

REVIEWERS' COMMENTS:

Reviewer #2 (Remarks to the Author):

The authors have tried to thoroughly address most of the remaining comments.

I would encourage to add the additional analyses in the Suppl Information.